# S100 Calcium-Binding Protein P Secreted from Megakaryocytes Promotes Osteoclast Maturation

**DOI:** 10.3390/ijms22116129

**Published:** 2021-06-07

**Authors:** Seung-Hoon Lee, Hye Jung Ihn, Eui Kyun Park, Jung-Eun Kim

**Affiliations:** 1Department of Molecular Medicine, School of Medicine, Kyungpook National University, Daegu 41944, Korea; jsat1234@naver.com; 2BK21 Four KNU Convergence Educational Program of Biomedical Sciences for Creative Future Talents, Department of Biomedical Science, Kyungpook National University, Daegu 41944, Korea; 3Cell and Matrix Research Institute, Kyungpook National University, Daegu 41944, Korea; hjpihn@hanmail.net; 4Department of Oral Pathology and Regenerative Medicine, School of Dentistry, IHBR, Kyungpook National University, Daegu 41944, Korea; epark@knu.ac.kr

**Keywords:** S100P, megakaryocyte, osteoclastogenesis

## Abstract

Megakaryocytes (MKs) differentiate from hematopoietic stem cells and produce platelets at the final stage of differentiation. MKs directly interact with bone cells during bone remodeling. However, whether MKs are involved in regulating bone metabolism through indirect regulatory effects on bone cells is unclear. Here, we observed increased osteoclast differentiation of bone marrow-derived macrophages (BMMs) cultured in MK-cultured conditioned medium (MK CM), suggesting that this medium contains factors secreted from MKs that affect osteoclastogenesis. To identify the MK-secreted factor, DNA microarray analysis of the human leukemia cell line K562 and MKs was performed, and S100 calcium-binding protein P (S100P) was selected as a candidate gene affecting osteoclast differentiation. S100P was more highly expressed in MKs than in K562 cells, and showed higher levels in MK CM than in K562-cultured conditioned medium. In BMMs cultured in the presence of recombinant human S100P protein, osteoclast differentiation was promoted and marker gene expression was increased. The resorption area was significantly larger in S100P protein-treated osteoclasts, demonstrating enhanced resorption activity. Overall, S100P secreted from MKs promotes osteoclast differentiation and resorption activity, suggesting that MKs indirectly regulate osteoclast differentiation and activity through the paracrine action of S100P.

## 1. Introduction

Megakaryocytes (MKs) are cells with large lobulated nuclei that differentiate from hematopoietic stem cells in the bone marrow [1,2]. The primary function of MKs is to release and maintain platelets, which are essential for blood coagulation [3]. During development, MKs are also found in the liver and spleen [4]. The primary signal for MK production is thrombopoietin, which binds to the c-Mpl receptor and concentration-dependently affects the proliferation and maturation of MK progenitor cells [5]. Other signals include cytokines and chemokines [6]. Interestingly, several studies have shown that MKs and platelets are involved in bone tissue healing. When bone tissue undergoes fractures, platelets move to the fracture site to minimize bleeding and contribute to the healing process by secreting growth factors and cytokines, indicating that platelets are required to mitigate the inflammatory response at the fracture site and to heal the bone fracture [7]. Additionally, clinical studies showed that platelet-rich plasma increases the healing rate and reduces the healing duration and infection incidence [8,9]. However, the specific MKs and platelets directly involved in bone metabolism are unclear. A direct interaction has been detected between MKs and osteoblasts, with MKs increasing osteoblast proliferation and bone mass, suggesting an anabolic role for MKs in skeletal homeostasis [10,11]. A recent study showed that MK-cultured conditioned medium (MK CM) affected bone cell differentiation in vitro and bone mass in ovariectomized mice, suggesting that MK-secreted factors play a role in bone metabolism [12]. Although some studies have focused on the indirect effects of MKs on bone cells or MK-secreted factors involved in bone metabolism [12,13,14,15,16], these effects and factors still remain unclear.

S100 calcium-binding protein P (S100P) is a member of the S100 family of proteins, which contains helix-loop-helix (EF-hand) motifs [17], and was first discovered and purified from the human placenta [18,19]. S100P is expressed in several healthy tissues, including the lungs, heart, kidneys, and bone marrow [20,21], as well as in tumor tissues, including those of pancreatic and breast cancers [22,23]. S100P is involved in diverse biological processes, including cellular calcium signaling and cancer progression [23,24,25]. However, its role in bone metabolism remains unknown.

In this study, we performed DNA microarray analysis of fully differentiated MKs to determine the MK-secreted factors affecting bone cell differentiation and identified the MK-secreted factor S100P to be responsible. Compared with control cells, S100P mRNA expression was significantly increased in MKs, and S100P protein was detected in MK CM. Additionally, the differentiation of mouse bone marrow macrophages (BMMs) into mature osteoclast-like cells was accelerated, and the bone-resorbing function of mature osteoclast-like cells was increased in the presence of S100P protein. Based on these findings, MK-secreted S100P promotes osteoclast differentiation and bone resorption.

## 2. Results

### 2.1. Osteoclast Differentiation in MK-Cultured CM

K562 cells were differentiated into mature MKs in phorbol 12-myristate 13-acetate (PMA)-supplemented medium for 7 days [26]; differentiation was determined by morphological changes, such as an increased nuclear/cytoplasmic ratio, large and multilobulated nuclei, and strong adhesion. Based on several reports that MKs may indirectly contribute to skeletal homeostasis [12,27], we investigated the effects of MK CM on osteoclast differentiation. Using tartrate-resistant acid phosphatase (TRAP) staining, TRAP-positive osteoclasts can be distinguished from TRAP-negative MKs [28]. The viability of BMMs cultured with K562-cultured conditioned medium (K CM) or MK CM, in either a dose-dependent or time-dependent manner, was not altered compared to that of BMMs cultured with control medium without CM (non-CM) (Appendix A). To determine the time point for differentiation, BMMs were cultured with 50% mixture of CM for 3, 4, and 5 days. TRAP-positive multinucleated cells (MNCs) were highly increased in MK CM on days 4 and 5 of differentiation compared to that in K CM and non-CM (Appendix A). The numbers of TRAP-positive MNCs were low on day 3 of osteoclast differentiation, indicating that a 3-day culture is insufficient for osteoclast differentiation (Appendix A). Moreover, TRAP-positive MNCs were significantly increased in BMMs cultured with 50% mixture of MK CM compared to that in BMMs cultured with 10% and 25% mixture (Appendix A). Finally, we decided to observe the effects of MK CM on osteoclast differentiation under 50% mixture of CM for 4 days of differentiation. TRAP staining showed that MK CM caused higher rates of BMM differentiation into osteoclast-like cells than did K CM and non-CM on day 4 of osteoclast differentiation (Figure 1A). Additionally, the number of TRAP-positive MNCs was significantly higher in MK CM-treated osteoclast-like cells than in non-CM- and K CM-treated osteoclast-like cells (Figure 1B). The expression of the osteoclast differentiation marker genes, cathepsin K, TRAP, and NFATc1, was significantly higher in MK CM-treated osteoclast-like cells than in non-CM- and K CM-treated osteoclast-like cells (Figure 1C). These results indicate that MK CM contains secreted factors that promote osteoclast differentiation.

### 2.2. Differential Gene Expression by Maturation of K562 Cells into MKs

Microarray analysis was performed to determine the differential gene expression profile for the maturation of K562 cells into MKs. Of the 44,629 genes compared using the Affymetrix GeneChip Human Gene 2.0 ST array, the expression of 955 genes was significantly changed, with an expression fold change of >2. Among these genes, 67 were secretory genes, and the expression of 28 was upregulated during K562 differentiation into MKs (Table 1). We analyzed 28 significantly upregulated genes based on GO biological process and molecular function provided by the Microarray analysis program (ExDEGA version 1.6.5) and literature reviews. Finally, we selected the S100P gene as a candidate MK-secreted factor affecting bone cell differentiation.

Comparison of mRNA expression in MKs and K562 cells by reverse transcription-polymerase chain reaction (RT-PCR) and quantitative PCR (qPCR) showed that S100P mRNA expression was significantly higher in MKs than in K562 cells (Figure 2A,B). A higher amount of S100P protein was secreted from MKs in MK CM than from those in K CM, as determined by enzyme-linked immunosorbent assay (ELISA) (Figure 2C).

### 2.3. Increased Osteoclast Differentiation and Bone-Resorbing Activity following S100P Treatment

S100P is expressed in various normal and tumor tissues and is involved in diverse biological functions [21]. However, S100P has rarely been studied in bone tissues and cells. Here, we investigated whether the candidate MK-secreted factor S100P, selected by microarray analysis, plays a role in osteoclast differentiation. Several studies have investigated the effect of recombinant S100P on cancer cell proliferation and survival, and an S100P concentration of 10–100 nM or more was found to show a significant effect [22,29]. Although the level of S100P secreted from MKs was approximately 50 pM (Figure 2C), this concentration can slightly vary, depending on the number or condition of MKs in vitro. Therefore, based on previous reports, recombinant human S100P protein was treated at a higher concentration than that of MK CM. Recombinant human S100P protein increased BMM differentiation into osteoclast-like cells at concentrations of 5, 10, and 20 nM, as determined by TRAP staining (Figure 3A). The number of TRAP-positive MNCs was significantly increased in the presence of recombinant human S100P protein (Figure 3B). The average size of TRAP-positive MNCs was similar among the 5, 10, and 20 nM S100P treatments (3123.54 ± 793.83 μm^2^, 3506.39 ± 167.36 μm^2^, and 3979.44 ± 640.11 μm^2^, respectively), indicating no significant difference in the size (*p* = 0.289 at 10 nM and *p* = 0.455 at 20 nM versus 5 nM). The expression of cathepsin K and TRAP was significantly increased at concentrations of 5 and 10 nM S100P, and NFATc1 expression showed a significant increase in osteoclast-like cells treated with 10 nM recombinant human S100P protein (Figure 3C). The control was treated with the same solvent containing 25 mM Tris-HCl (pH 7.3), 100 mM glycine, and 10% glycerol, which the human S100P protein was supplied in. These results indicate that S100P is secreted by MKs to promote osteoclast differentiation.

To determine the effect of S100P on bone-resorbing activity, BMMs were treated with or without 5 nM S100P protein for 5 days, and the resorption pit area of dentin slices was measured. Based on previous reports [30,31], both round-shaped single pits and crescent-shaped pits were identified as resorption pits. The toluidine blue-stained resorption pit area was larger (Figure 4A), and relative bone resorption area was significantly increased in S100P-treated osteoclast-like cells compared with that in control cells (Figure 4B). These results indicate that osteoclast activity for bone resorption is enhanced by S100P protein secreted from MKs.

## 3. Discussion

MKs are platelet-forming cells in the bone marrow involved in remodeling [3]. MKs themselves perform several actions in various tissues in vivo [3,4]. Although the role of MKs in the skeletal system has not been actively investigated, several studies have revealed a direct interaction and function between MKs and bone cells in regulating skeletal homeostasis. Most previous studies showed that MKs increase bone mass by upregulating bone formation in osteoblasts and inhibiting bone resorption by osteoclasts [11,32]. Mice deficient in GATA-1 or NF-E2, which are major terminal differentiation factors of MKs [33], exhibited decreased platelet production and increased number of immature MKs, leading to induction of a potential interaction between MKs and osteoblasts to increase bone mass [10]. However, the reduction in osteoclast formation and function by MKs is indirect rather than direct, because the level of MK-expressing osteoprotegerin was high in MK CM, thereby inhibiting osteoclast development [16]. In this study, we identified more upregulated genes (relative gene expression fold change >2 and *p*-value < 0.05) in MKs than in K562 cells. The expression of osteoprotegerin (TNFRSF11B, tumor necrosis factor receptor superfamily member 11 B) was similar in MKs and K562 cells, showing a fold change of 0.945 and *p*-value of 0.399 (data not shown). This result indicates that osteoprotegerin expression was not altered in MKs compared to that in K562 cells in the culture condition used in this study. Interestingly, recent studies have focused on the controversial function of MKs, demonstrating that their impact on the skeletal system may vary under different conditions. With aging, the number of MKs increases, and the direct effect of MKs on promoting bone formation in osteoblasts is suppressed [34]. Increased MKs in aged mice activate RANKL expression and signaling, thereby promoting osteoclastogenesis and bone resorption activity [27]. Another study showed that MK CM regulates osteoblast and osteoclast differentiation and increases bone mass in ovariectomized mouse models, although the MK-secreted factors in MK CM have not been identified [12]. Moreover, the role of MKs in bone metastasis remains unclear. Some studies on bone metastasis reported a protective role of MKs, whereas others suggested a progressive role of MKs [35,36,37]. Particularly, studies have shown that MK-releasing factors stimulate osteolytic bone metastasis [36]. These results suggest that bone homeostasis is regulated not only by MKs but also by MK-secreted factors, suggesting a direct and indirect role of MKs in the skeletal system. However, studies on the MK-secreted factors involved in regulating bone metabolism are lacking. Therefore, we selected a candidate protein that can affect osteoclast differentiation among the factors secreted by MKs, and confirmed its function in osteoclasts.

Microarray analysis was performed to compare the secretory factors affecting osteoclasts derived from MKs and K562 cells, and S100P was selected as a candidate factor. The S100 family of calcium-binding proteins contains approximately 16 members such as S100A, S100B, S100C, and S100P. Interestingly, there are several reports that some of the S100 family of proteins play a role in bone metabolism. S100A4, S100A8/A9, and S100A12 have been reported to facilitate osteoclast differentiation and bone resorption [38,39,40]. Due to these reports, S100P could be a potential candidate to function in osteoclastogenesis. In addition, several studies have shown that S100P plays an important role in cancer development. S100P is expressed in various organs, including the pancreas, breast, and prostate, and shows highly increased expression in most cancers, suggesting that S100P can be used as a prognostic marker [20,22,25]. S100P binds to the receptor for advanced glycation end-products (RAGE) and activates cellular signaling to mediate tumor growth and metastasis [41]. The interaction between S100P and RAGE specifically activates the nuclear factor-kappa B (NF-κB) signaling pathway, which increases resistance to therapeutic agents by promoting cancer cell survival and proliferation [41,42]. NF-κB signaling is essential for regulating osteoclastogenesis [43]. Interestingly, in RAGE-deficient mice, impaired NF-κB signaling increases bone mass by reducing osteoclast maturation and activity [44]. However, these results do not support the fact that S100P is involved in the regulation of osteoclastogenesis. We found that the mRNA expression of S100P was increased 4–6-fold by the maturation of K562 cells into MKs. Additionally, the concentration of S100P protein secreted into MK CM was 4-fold higher than that into K CM, although protein expression was not confirmed because antibodies suitable for Western blotting were not available. These results indicate that MKs show increased expression and secretion of S100P. To determine the effect of MK-secreted S100P, BMMs were differentiated into mature osteoclast-like cells in the presence of S100P protein. S100P is encoded by the *S100P* gene in humans [18], but information on mouse S100P is insufficient. House mice contain S100P (GenBank _BC038329.1) or a predicted gene, 40318 (Official Symbol _Gm40318), as an S100P pseudogene; however, they show no nucleotide and protein sequence similarity to human S100P. Arumugam et al. [41] investigated the biological effects of human S100P protein on the proliferation and survival of NIH3T3 cells, which are fibroblasts isolated from mouse embryos. They demonstrated that human S100P protein activates NF-κB and RAGE, ultimately stimulating NIH3T3 cell proliferation and survival. NF-κB shares nucleotide and protein sequence identities of 84% and 86%, respectively, between humans and mice, whereas RAGE shares nucleotide and protein sequence identities of 79% and 78%, respectively, between these species [45,46]. In addition, it has been reported that RAGE is expressed in mouse BMMs, preosteoclasts, and mature osteoclasts [44]. We also confirmed NF-κB and RAGE mRNA expression in mouse BMMs (data not shown). Therefore, human recombinant S100P protein was used in this study. However, further studies are still needed to comprehensively explore the role of human recombinant S100P in human osteoclast progenitors. Interestingly, although human recombinant S100P increased osteoclast differentiation, a high dose of S100P did not enhance the expression of marker genes for osteoclast differentiation (Figure 3). A high dose of S100P may cause side effects that reduce marker gene expression or arrest osteoclast differentiation and trigger intracellular signaling responsible for osteoclast apoptosis at the observed time point. Further studies are needed to clarify this alteration. In this study, our results indicated that S100P promoted osteoclast differentiation and increased bone resorption activity. Unfortunately, we could not confirm the action of MK-secreted S100P in osteoclastogenesis, as S100P-neutralizing antibodies are not available. Additional studies are needed to generate a S100P-neutralizing antibody or blocker of the S100P/RAGE interaction and to determine the solid role of S100P in osteoclastogenesis. As blocking of the S100P/RAGE interaction is clinically beneficial for cancer therapy, blocking of S100P function may also be helpful in inhibiting osteoclast differentiation and activity in senile osteoporosis or osteolysis caused by bone metastasis of cancer cells. Moreover, the coupling action of osteoblasts and osteoclasts is important for bone remodeling [47,48]. Further studies are necessary to understand whether S100P plays a role in osteoblastogenesis and to determine the physiological role of S100P in the coupling action of osteoblasts and osteoclasts.

Initially, MKs were predicted to be unique cells with distinct morphological features that facilitate blood clotting by forming platelets. However, MKs have been shown to play several roles in various tissues, either directly or indirectly, through numerous factors and signaling pathways. We demonstrated that the MK-derived factor S100P, rather than MKs themselves, promoted osteoclast differentiation and bone resorption activity.

## 4. Materials and Methods

### 4.1. Cell Culture and Collection of CM

The human leukemic cell line K562 (ATCC, Manassas, VA, USA) was used to generate MKs. K562 cells were cultured in RPMI 1640 medium (HyClone, Logan, UT, USA) containing 10% fetal bovine serum (FBS; HyClone), 100 U/mL penicillin, and 100 μg/mL streptomycin (Gibco, Grand Island, NY, USA). Cells were treated with 1 nM PMA (Sigma-Aldrich, St. Louis, MO, USA) for 7 days to induce differentiation into MKs [26]. The CM from MKs or K562 control cells was collected after a 24 h incubation period in fresh serum-free RPMI 1640 medium without PMA. To remove the MKs or cell debris from the collected CM, centrifugation was performed at 80× *g* for 5 min. BMMs were isolated from the femurs and tibias of 8-week-old C57BL/6 mice by centrifugation and then cultured in α-MEM supplemented with 10% FBS, 100 U/mL penicillin, and 100 μg/mL streptomycin for 24 h at 37 °C in an atmosphere of 5% CO_2_. BMMs were seeded at a density of 1.0 × 10^5^ cells/well in 24-well culture plates for RNA extraction and 2.0 × 10^4^ cells/well in 96-well culture plates for TRAP staining or on dentin slices for the resorption pit assay in the presence of macrophage colony-stimulating factor (M-CSF; R&D Systems, Minneapolis, MN, USA). The cells were treated with 20 ng/mL M-CSF and 20 ng/mL RANKL (R&D Systems) for 4 days in a 50% mixture of CM or by adding recombinant human S100P protein (LSBio, Seattle, WA, USA) at concentrations of 5, 10, and 20 nM to induce osteoclast differentiation. Osteoclast differentiation was detected by TRAP staining using a TRAP staining kit (Cosmo Bio Co., Ltd., Tokyo, Japan) as described previously [49]. TRAP-positive MNCs with more than three nuclei were considered as differentiated osteoclast-like cells. The size of TRAP-positive MNCs was measured using the i-Solution image analysis program (IMT i-Solution, Daejeon, Korea). For the resorption pit assay, the cells were cultured on dentin slices for 5 days in the presence of S100P protein. The dentin slices were stained with 0.1% toluidine solution, and the area of the pit was measured using the i-Solution image analysis program. To observe cell proliferation, BMMs were cultured in a 10%, 25%, and 50% mixture of CM for 6, 12, and 24 h. Then, cell viability was determined using the 3-(4,5-dimethylthiazol-2-yl)-2,5-diphenyltetrazolium bromide assay.

### 4.2. RNA Extraction, RT-PCR, and qPCR

Total RNA was extracted from cultured cells using TRIzol reagent (Sigma-Aldrich, St. Louis, MO, USA). Reverse transcription was performed using reverse transcriptase premix (ELPIS-Biotech, Daejeon, Korea), and 1 μg of total RNA was used to generate cDNA. PCR was performed to amplify each gene under the following conditions: denaturation at 94 °C for 2 min; 30 cycles of amplification with denaturation at 94 °C for 30 s, annealing at 60–62 °C for 30 s, and extension at 72 °C for 1 min; and final extension at 72 °C for 10 min. The products were separated on 1.5% agarose gels and visualized by ethidium bromide staining. qPCR was performed using Power SYBR Green PCR Master Mix (Applied Biosystems, Foster City, CA, USA) on a StepOnePlus Real-Time PCR System (Applied Biosystems, Foster City, CA, USA) as follows: initial denaturation at 95 °C for 5 min; 45 cycles of amplification with denaturation at 95 °C for 30 s, annealing at 64 °C for 30 s, and extension at 72 °C for 60 s; one cycle for melting curve analysis at 95 °C for 5 s, 65 °C for 1 min, and 97 °C continuous; and a final cooling step at 40 °C for 30 s. The results were analyzed using the comparative cycle threshold (*C_T_*) method. The primer sets used for RT-PCR and qPCR are listed in Table 2.

### 4.3. Affymetrix cDNA Microarray and Data Analysis

cDNA synthesized using total RNA from K562 cells and MKs was used to conduct an Affymetrix GeneChip Human Gene 2.0 ST array (D&P Biotech, Inc., Daegu, Korea) as previously described [50]. Microarray data were analyzed using ExDEGA version 1.6.5 (EBIOGEN, Inc., Seoul, Korea). A relative gene expression fold change >2, normalized data (log2) > 4, and *p*-value < 0.05 were considered to represent upregulation or downregulation.

### 4.4. ELISA

The concentration of S100P protein in K CM and MK CM was measured using the Human S100 calcium-binding protein P ELISA kit (MyBioSource, Inc., San Diego, CA, USA) according to the manufacturer’s instructions. Briefly, CM was centrifuged at 1000× *g* for 20 min to collect the supernatant, which was immediately used in the assay. CM (100 μL) was added to pre-coated wells with an S100P-specific antibody and incubated at 37 °C for 90 min. After removing the liquid from each well, 100 μL of a biotin-conjugated antibody specific for S100P was added and incubated for 45 min at 37 °C. After adding 100 μL of horseradish peroxidase-conjugated avidin, the plate was incubated for 45 min at 37 °C. After removing the solution, the wells were washed five times for 1 min each time with 300 μL of washing solution, and the remaining liquid was removed. The samples were reacted with 90 μL of 3,3′,5,5′-tetramethylbenzidine substrate for 20 min at 37 °C in the dark to detect S100P binding, and the color reaction was blocked by adding 50 μL of stop solution. The optical density of each well was determined spectrophotometrically at a wavelength of 450 nm.

### 4.5. Statistical Analysis

Cell culture experiments were conducted on at least three independent occasions, and each experiment was performed in duplicate. All data are presented as the mean ± standard error of the mean and compared by Student’s *t*-test. A *p*-value < 0.05 was considered statistically significant.

## Figures and Tables

**Figure 1 ijms-22-06129-f001:**
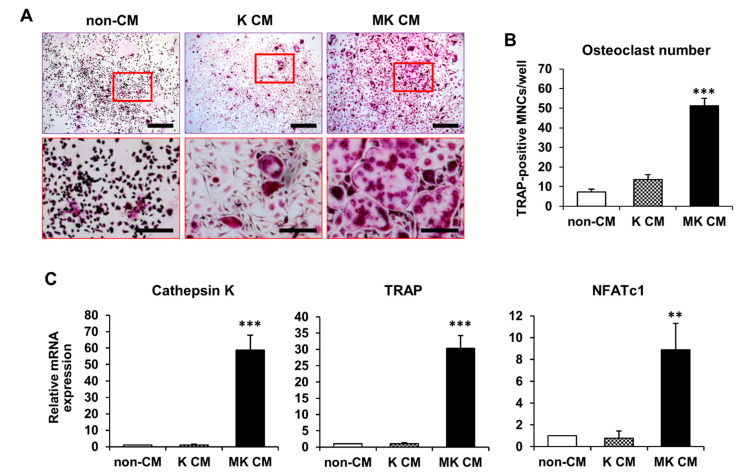
Effect of MK CM on the differentiation of bone marrow-derived macrophages (BMMs) into mature osteoclast-like cells. (**A**) BMMs were cultured in osteoclast differentiation medium containing M-CSF (20 ng/mL) and RANKL (20 ng/mL) in the presence of K562-cultured CM (K CM) or MK CM. The non-CM control group was cultured in osteoclast differentiation medium mixed with 50% of RPMI1640. The cells were stained with TRAP to examine osteoclast differentiation. Scale bar, 500 μm. High-magnifications of the red rectangles in the upper panels are shown in the lower panels. Scale bar, 200 μm. (**B**) Number of TRAP-positive multinucleated cells (MNCs) with three or more nuclei was counted. ***, *p* < 0.001 versus non-CM. (**C**) mRNA expression of cathepsin K, TRAP, and NFATc1 was evaluated by qPCR during osteoclast differentiation. The relative mRNA expression level was plotted against gene expression levels in non-CM, which were set to 1.0. The results are representative of three independent experiments. **, *p* < 0.01 and ***, *p* < 0.001 versus non-CM.

**Figure 2 ijms-22-06129-f002:**
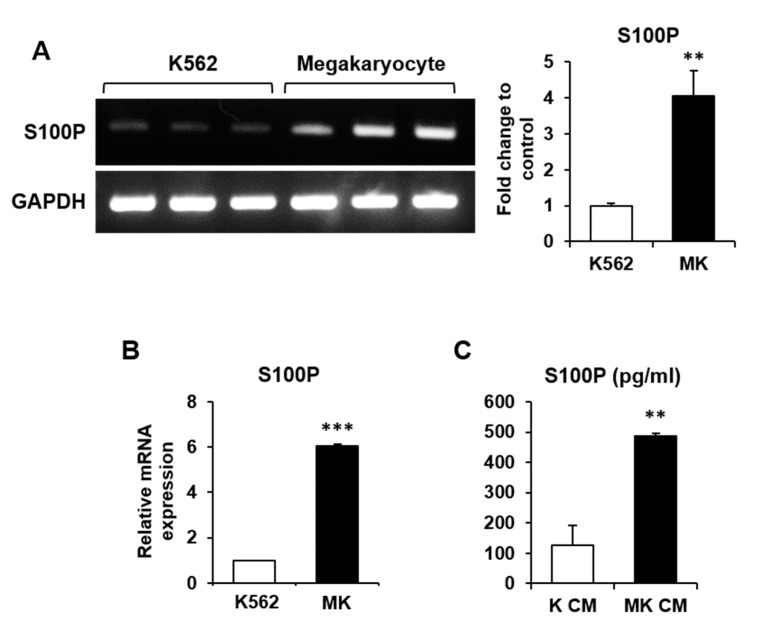
Validation of S100P expression in megakaryocytes (MKs) and MK-cultured conditioned medium (MK CM). (**A**,**B**) S100P mRNA expression in K562 and MKs detected by RT-PCR (**A**) and qPCR (**B**). The intensity of individual bands after RT-PCR (**A**) was determined using ImageJ software (NIH, Bethesda, MD, USA), and the data were normalized to the expression of Gapdh and calculated as a fold change relative to that in K562 cells, which was set to 1.0. **, *p* < 0.01. The relative mRNA expression level (**B**) was plotted against gene expression levels in K562 cells, which were set to 1.0. ***, *p* < 0.001. (**C**) Concentration of S100P protein in K CM and MK CM was determined by ELISA. **, *p* < 0.01 versus K CM.

**Figure 3 ijms-22-06129-f003:**
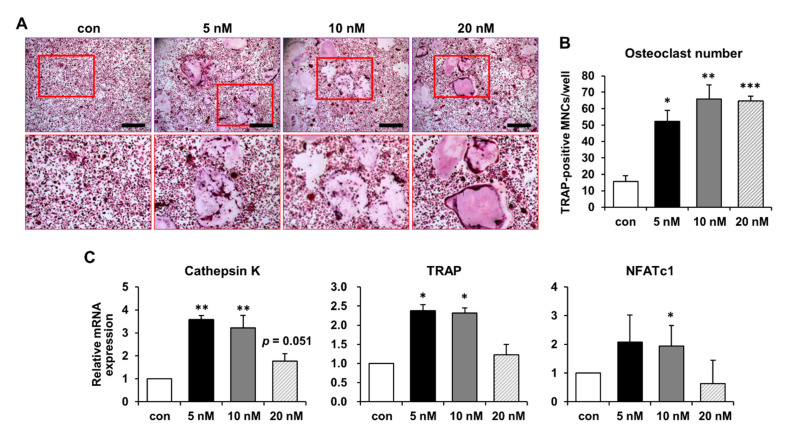
Effect of recombinant human S100P protein on the differentiation of bone marrow-derived macrophages (BMMs) into mature osteoclast-like cells. (**A**) BMMs were cultured in osteoclast differentiation medium containing M-CSF (20 ng/mL) and RANKL (20 ng/mL) in the presence of S100P at concentrations of 5, 10, and 20 nM. The cells were stained with TRAP to examine osteoclast differentiation. Scale bar, 500 μm. High magnifications of the red rectangles in the upper panels are shown in the lower panels. (**B**) Number of TRAP-positive multinucleated cells (MNCs) with three or more nuclei was counted. *, *p* < 0.05; **, *p* < 0.01; ***, and *p* < 0.001 versus the control (con). (**C**) mRNA expression of cathepsin K, TRAP, and NFATc1 was evaluated by qPCR after treatment with 5, 10, and 20 nM S100P protein during osteoclast differentiation. The relative mRNA expression level was plotted against gene expression levels in the control (con), which were set to 1.0. The results are representative of three independent experiments. *, *p* < 0.05 and **, *p* < 0.01 versus con.

**Figure 4 ijms-22-06129-f004:**
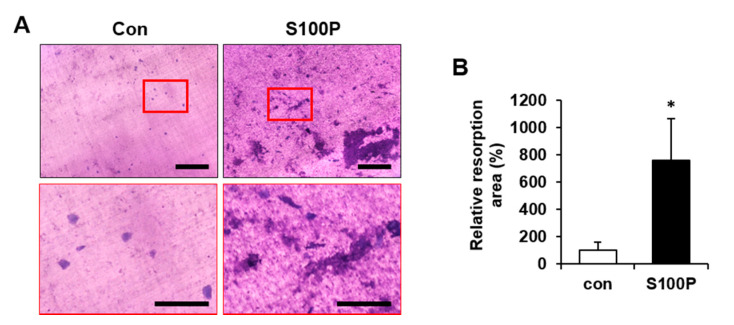
Effect of recombinant human S100P protein on bone resorptive activity. (**A**) Bone marrow-derived macrophages (BMMs) were cultured in osteoclast differentiation medium containing M-CSF (20 ng/mL) and RANKL (20 ng/mL) in the presence of 5 nM S100P. After 5 days of differentiation, dentin slices were stained with toluidine blue solution to observe the resorption pit, and representative images are shown. Scale bar, 500 μm. High-magnification images of the red rectangles in the upper panels are shown in the lower panels. Scale bar, 200 μm. (**B**) Quantification of the percentage of bone resorption area per total dentin slice using the i-Solution image analysis program. The resorption pit area is relative to that of the dentin slices of cultured BMMs without S100P, which was set to 100%. *, *p* < 0.05.

**Table 1 ijms-22-06129-t001:** Upregulated secretory genes in response to K562 differentiation into MKs.

Gene Accession	Gene Name (Gene Symbol)	Fold Change	*p* Value
NM_005139	annexin A3 (ANXA3)	4.458	0.000
NM_001306129	fibronectin 1 (FN1)	3.972	0.000
NM_001062	transcobalamin I (vitamin B12 binding protein, R binder family) (TCN1)	3.491	0.001
NM_000239	lysozyme (LYZ)	3.460	0.007
NM_003054	solute carrier family 18 (vesicular monoamine transporter), member 2 (SLC18A2)	3.364	0.000
NM_001024845	solute carrier family 6 (neurotransmitter transporter, glycine), member 9 (SLC6A9)	2.715	0.005
NM_001243245	proteoglycan 2, bone marrow (natural killer cell activator, eosinophil granule major basic protein) (PRG2)	2.573	0.000
NM_001292045	neuromedin U (NMU)	2.469	0.000
NM_002031	fyn-related Src family tyrosine kinase (FRK)	2.465	0.000
NM_000632	integrin, alpha M (complement component 3 receptor 3 subunit) (ITGAM)	2.438	0.000
NM_001282386	isocitrate dehydrogenase 1 (NADP+) (IDH1)	2.377	0.000
NM_021109	thymosin beta 4, X-linked (TMSB4X)	2.374	0.011
NM_000132	coagulation factor VIII, procoagulant component (F8)	2.366	0.022
NM_000405	GM2 ganglioside activator (GM2A)	2.322	0.001
NM_002562	purinergic receptor P2X, ligand gated ion channel, 7 (P2RX7)	2.313	0.003
NM_001303499	calponin 2 (CNN2)	2.295	0.001
NM_005980	S100 calcium binding protein P (S100P)	2.245	0.008
NM_001042402	N-acylethanolamine acid amidase (NAAA)	2.194	0.000
NM_001063	transferrin (TF)	2.194	0.001
NM_004472	forkhead box D1 (FOXD1)	2.130	0.000
NM_000189	hexokinase 2 (HK2)	2.124	0.000
NM_001159629	solute carrier family 27 (fatty acid transporter), member 2 (SLC27A2)	2.123	0.000
NM_001098503	protein tyrosine phosphatase, receptor type, J (PTPRJ)	2.113	0.000
NM_001145031	plasminogen activator, urokinase (PLAU)	2.102	0.002
NM_001025366	vascular endothelial growth factor A (VEGFA)	2.065	0.000
NM_001078175	solute carrier family 29 (equilibrative nucleoside transporter), member 1 (SLC29A1)	2.062	0.000
NM_001244984	sel-1 suppressor of lin-12-like (C. elegans) (SEL1L)	2.044	0.000
NM_004335	bone marrow stromal cell antigen 2 (BST2)	2.031	0.002

**Table 2 ijms-22-06129-t002:** Primer sequences for RT-PCR and qPCR.

Name	Product	Primer Sequences	
RT-PCR			
S100P	97 bp	F	5’-GAG AAG GAG CTA CCA GGC TTC-3’
		R	5’-TCC ACC TGG GCA TCT CCA TT-3’
Gapdh	240 bp	F	5’-TGA TGA CAT CAA GAA GGT GGT GAA G-3’
		R	5’-TCC TTG GAG GCC ATG TAG GCC AT-3’
qPCR			
S100P	97 bp	F	5’- GAG AAG GAG CTA CCA GGC TTC-3’
		R	5’- TCC ACC TGG GCA TCT CCA TT-3’
CtsK	178 bp	F	5’-CAG AAC GGA GGC ATT GAC-3’
		R	5’-CGA TGG ACA CAG AGA TGG-3’
Trap	197 bp	F	5’-GCA GCC AAG GAG GAC TAC-3’
		R	5’-CCC ACT CAG CAC ATA GCC-3’
Nfatc1	161 bp	F	5’-TCT TCC GAG TTC ACA TCC-3’
		R	5’-ACA GCA CCA TCT TCT TCC-3’
Gapdh	126 bp	F	5’-GCA TCT CCC TCA CAA TTT CCA-3’
		R	5’-GTG CAG CGA ACT TTA TTG ATG G-3’

S100p, S100 calcium-binding protein P; CtsK, cathepsin K; Trap, tartrate-resistant acid phosphatase; Nfatc1, nuclear factor of activated T-cell cytoplasmic 1; Gapdh, glyceraldehyde 3-phosphate dehydrogenase; F, forward; R, reverse.

## Data Availability

Data can be obtained from the corresponding author.

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
