# Peer review of "S100 Calcium-Binding Protein P Secreted from Megakaryocytes Promotes Osteoclast Maturation"

_ijms, 2021, doi:10.3390/ijms22116129_

Round 1

Reviewer 1 Report

The authors have extensively revised their original manuscript. Most of my concerns have been addressed. Accordingly, the conclusion that human S100P accelerates mouse osteoclast differentiation has been consolidated.

Human monocytes are commercially available. When you examine the effect of human S100P on the osteoclastogenesis from the monocytes, You will be able to extend your conclusion to human osteoclasts. This additional experiments would heighten the value of your work.

Reviewer 2 Report

Megakaryocytes (MKs) have been reported to be involved in bone tissue healing, but the details including the mechanisms and/or molecular factors which mediate indirect effects of MKs on bone cells are not yet clear. In this paper, Lee et al. challenged to identify the factor which is produced by MKs and affects osteoclast differentiation, and identified S100P protein in condition medium of MK culture by DNA microarray analysis. Indeed, S100P was expressed in MKs, secreted into MK culture, and commercially available recombinant human S100P enhanced osteoclast differentiation and resorption activities.

This is an interesting paper. I was enjoying reading this manuscript. Identifying the molecule which could mediate MK-osteoclast interaction will contribute to understanding complicated bone homeostasis. I have several questions and requests, and I hope the authors will address in revision.

Major,

It is essential to show whether blocking S100P in MK condition medium wipe out (or diminish) the effect on osteoclast differentiation. Although the authors described in Discussion that there is no S100P neutralizing antibodies, I have searched PubMed and found that there is a paper which developed function-blocking anti-S100P monoclonal antibodies (Oncogenesis, S Dakhel et al. 2014, doi: 10.1038/oncsis.2014.7). Although these mAbs are anti-human S100P, it is worth to test them given that recombinant human S100P showed enhancement of osteoclast differentiation. Please test neutralization of S100P with these mAbs. I am very curious the results.

Minor,

Basically, this paper is well written. But it is not clear how or why the authors chose S100P among 28 genes. What criteria did the authors use to choose S100P? Or is S100P just the first gene to be examined as MK-secreting factor, and the authors keep continuing investigation of left 27 genes? Is S100P the sole factor which is derived from MKs and affects osteoclasts among 28 genes? Or are there possibilities that left 27 genes affect bone cells?

In figure 3, 20 nM of rhS100P did not show enhancement on induction of CatK, TRAP, and NFATc1 although this concentration enhanced osteoclast formation. Please add explain of this point.

Line 148-149: this sentence is not clear. Does this mean that the authors used solvent of recombinant protein as a control?

Round 2

Reviewer 2 Report

The authors answered all my questions. I'm satisfied with this ver. of paper.

This manuscript is a resubmission of an earlier submission. The following is a list of the peer review reports and author responses from that submission.

Round 1

Reviewer 1 Report

In the manuscript entitled: “A secreted protein of megakaryocytes, S100 calcium binding protein P, promotes osteoclast maturation“ is investigated that a factor secreted by megakaryocytes promotes osteoclast formation and activity.  With qPCR was found that S100P was upregulated and with recombinant S100P protein they also found an enhanced osteoclast activity and formation. However, some points need to be clarified.

  1. The magnification of the micrographs of osteoclasts is very low which make it difficult to see if these cells are multinucleated. Especially in figure 1 many cells are trap positive MK cells and not osteoclasts.
  2. How can the difference in osteoclast number between 5 and 20 nM explained? Are osteoclasts in 20nM larger (with more nuclei per cell)? What about the mRNA expression of TRAP, Cathepsin K and NFATc1 with this concentration?
  3. Was the culture time only 4 days?
  4. I am not sure if the blue spot a real resorption pit. Normally pits are more round spots or trenches were formed by the osteoclasts but they do not look like this. Since the magnification is comparable to the osteoclast micrographs it seems no resorption to me.
  5. In line 168 is mentioned that MK CM regulates osteoblast and osteoclast differentiation. In this manuscript only osteoclast differentiation is investigated. It would be nice to do a co-culture with osteoblasts and (pre) osteoclasts to investigate a more physiological role of S100P on these cells.

Author Response

Please see the attachment. Thank you in advance.

Reviewer 2 Report

Lee et al. report the role of S100P secreted from megakaryocytes on osteoclast differentiation in this manuscript. They first asked whether megkaryocytes secrete a factor that affect osteoclastogenesis. For this, the authors isolated the culture media of megakaryocytes differentiated from human leukemic cell line, K562. Osteoclastogenesis from mouse BMM was increased by the addition of the conditioned media from megakaryocytes (Figure 1). This is confirmed by the enhanced expression of mRNA of NFATc1, cathepsin K, and TRAP. Next, they searched the unregulated genes during the differentiation of megakaryocytes using the GeneChip, and selected S100P as a candidate for the factor that increased osteoclastogenesis. The upregulation and secretion of S100P was confirmed by RT-PCR, qPCR, and ELISA (Figure 2). The addition of the recombinant human S100P protein to the culture increased the osteoclast differentiation (Figure 3). Finally, the recombinant protein increased the bone resorption of mouse BMM differentiated on a dentine slice (Figure 4). Based on these results, the authors concluded that megakaryocytes indirectly regulate osteoclast differentiation through the paracrine action of S100P. This is the first report that megakaryocytes indirectly increase osteoclastogenesis via S100P in vitro. Unfortunately, the experimental design of this study had serious flaws. The authors should include important previous works that reported the inhibitory effect of megakaryocytic on osteoclastogenesis in “Introduction”. I hope my comments below would be helpful to improve the manuscript.  

Major comments:

1.   Line 52. “However few studies have been conducted on…”

     There are several studies that examined the role of megaryocytes on osteoclastogenesis (Beeton, Bone, 39:985, 2006; Kacena, Bone, 39:991, 2006; Ciovacco, J Cell Biochem, 109:774, 2010; Kacena & Ciovacco, Adv Exp Med Biol, 658:31, 2010; Lee, Sic Rep, 10:2277, 2020).  All these works reported that megaryocytes inhibited osteoclastogenesis. The authors should cite these works in “Introduction”.   

2.   Fig.1A and 1B.

      The authors examined the effect of MK-CM on osteoclastogenesis at one concentration of MK-CM. A does-response curve is preferable to validate the conclusion. Because MK-CM increased the proliferation of BMM (Lee, 2020), the authors are recommended to test the effect of MK-CM and K-CM on the cell proliferation of BMM. Furthermore, a previous study showed that the number of differentiated osteoclasts depended on the culture period (Lee, 2020). Thus, time course study is needed to confirm your conclusion.

3.  Fig. 3A, 3B, 3C.

     It is curious that the number of osteoclasts generated by 20 nM S100P was smaller than that by 5 nM. The authors are recommended to show a dose-response curve of S100P on osteoclastogenesis.

    The S100P you used (LSBio) appears to contains additives such as NP-40, NA3VO4, or 100 mM glycine and 10% glycerol or  10mM glutathione, etc. Such additives might enhance osteoclastogenesis. Did you remove the additives by dialysis or gel filtration? Or did you add the same amount of such additives to the control culture? If so, please describe it in “Materials and Methods”.

4.  Line 162 - 163.  “…because the level of MK-expressing osteoprotegerin was high in MK CM, resulting in the inhibition of osteoclast development [26]”.

      Reference 26 showed that the high levels of osteoprotegerin in MK-CM was not the factor that caused the inhibition of osteoclastogenesis. Did you neutralize or remove osteoprotegerin from your MK-CM by pretreatments? If not, please describe the possible effect of osteoprotegerin in your MK-CM in “Discussion”.

Minor comments:

1.   Line 29. “demonstrating improved resorption activity”.

      Line 142. “ bone resorption was promoted by S100P protein secreted from MKs”

     The results in Figure 4 suggest that osteoclasts differentiated from mouse BMM on a dentine slice showed an increase in bone resorption in the presence of recombinant S100P. If you want to say “improve” or “promote”, you have to count the number of osteoclasts on a dentine slice and compare the ability of bone resorption such as resorption area/osteoclast. I think the results in Figure 4 merely indicate that the S100P  enhanced the number of osteoclast-like cells that have the ability to resorb bone.

2.    The researcher in bone biology traditionally use the term “osteoclast-like cells” for the multinucleated cells differentiated in vitro. The term “osteoclasts” is only used for the cells isolated from bone. The manuscript would be improved if the author could follow the convention throughout the text.

3.   The authors used the recombinant human S100P in their experiments. It will be useful for readers to explain why human recombinant worked on mouse BMM in “Discussion". Are there any differences in the protein sequence between human and mouse S100P?

Author Response

Reviewer #2

Lee et al. report the role of S100P secreted from megakaryocytes on osteoclast differentiation in this manuscript. They first asked whether megkaryocytes secrete a factor that affect osteoclastogenesis. For this, the authors isolated the culture media of megakaryocytes differentiated from human leukemic cell line, K562. Osteoclastogenesis from mouse BMM was increased by the addition of the conditioned media from megakaryocytes (Figure 1). This is confirmed by the enhanced expression of mRNA of NFATc1, cathepsin K, and TRAP. Next, they searched the unregulated genes during the differentiation of megakaryocytes using the GeneChip, and selected S100P as a candidate for the factor that increased osteoclastogenesis. The upregulation and secretion of S100P was confirmed by RT-PCR, qPCR, and ELISA (Figure 2). The addition of the recombinant human S100P protein to the culture increased the osteoclast differentiation (Figure 3). Finally, the recombinant protein increased the bone resorption of mouse BMM differentiated on a dentine slice (Figure 4). Based on these results, the authors concluded that megakaryocytes indirectly regulate osteoclast differentiation through the paracrine action of S100P. This is the first report that megakaryocytes indirectly increase osteoclastogenesis via S100P in vitro. Unfortunately, the experimental design of this study had serious flaws. The authors should include important previous works that reported the inhibitory effect of megakaryocytic on osteoclastogenesis in “Introduction”. I hope my comments below would be helpful to improve the manuscript.

Major comments:

  1. Line 52. “However few studies have been conducted on…”

There are several studies that examined the role of megaryocytes on osteoclastogenesis (Beeton, Bone, 39:985, 2006; Kacena, Bone, 39:991, 2006; Ciovacco, J Cell Biochem, 109:774, 2010; Kacena & Ciovacco, Adv Exp Med Biol, 658:31, 2010; Lee, Sic Rep, 10:2277, 2020).  All these works reported that megaryocytes inhibited osteoclastogenesis. The authors should cite these works in “Introduction”.   

>> We apologize for omitting important references. As per the reviewer’s suggestion, we have cited the references in the Introduction section (Lines 50-52) and added the following references [12-16] to the References section.

[12] Lee, Y.S.; Kwak, M.K.; Moon, S.A.; Choi, Y.J.; Baek, J.E.; Park, S.Y.; Kim, B.J.; Lee, S.H.; Koh, J.M. Regulation of bone metabolism by megakaryocytes in a paracrine manner. Sci. Rep. 2020, 10, 2277.

[13] Beeton, C.A.; Bord, S.; Ireland, D.; Compston, J.E. Osteoclast formation and bone resorption are inhibited by megakaryocytes. Bone 2006, 39, 985-990.

[14] Ciovacco, W.A.; Cheng, Y.H.; Horowitz, M.C.; Kacena, M.A. Immature and mature megakaryocytes enhance osteoblast proliferation and inhibit osteoclast formation. J. Cell. Biochem. 2010, 109, 774-781.

[15] Kacena, M.A.; Ciovacco, W.A. Megakaryocyte-bone cell interactions. Adv. Exp. Med. Biol. 2010, 658, 31-41.

[16] Kacena, M.A.; Nelson, T.; Clough, M.E.; Lee, S.K.; Lorenzo, J.A.; Gundberg, C.M.; Horowitz, M.C. Megakaryocyte-mediated inhibition of osteoclast development. Bone 2006, 39, 991-999.

  1. 1A and 1B.

The authors examined the effect of MK-CM on osteoclastogenesis at one concentration of MK-CM. A does-response curve is preferable to validate the conclusion. Because MK-CM increased the proliferation of BMM (Lee, 2020), the authors are recommended to test the effect of MK-CM and K-CM on the cell proliferation of BMM. Furthermore, a previous study showed that the number of differentiated osteoclasts depended on the culture period (Lee, 2020). Thus, time course study is needed to confirm your conclusion.

>> Thank you very much for the reviewer’s comments. As pointed out by the reviewer, we investigated the effect of MK CM on osteoclastogenesis and proliferation according to the MK CM dose. Because BMMs did not grow well in RPMI 1640 medium, which is a conditioned medium from MKs, BMMs were cultured for osteoclast proliferation and differentiation as 10%, 25%, and 50% mixtures with MK CM and α-MEM. The does-response effects of K CM and MK CM on the proliferation of BMMs have been added to the revised manuscript (Lines 262-264 in the Materials and Methods, Lines 81-83 in the Results, and Figure S1). There was no significant difference in BMM proliferation according to the dose of CM, as determined by MTT assay. By the way, unfortunately, we could not obtain a sufficient amount of CM to perform additional experiments for dose-response during osteoclast differentiation. For osteoclast differentiation, 50% mixtures with MK CM and α-MEM was used (Figure 1).

In this study, we performed cell culture to observe osteoclast differentiation for 4 days after RANKL treatment. As pointed out by the reviewer, we conducted 2 day culture after RANKL treatment to observe the time-course effect of K CM and MK CM on osteoclast differentiation of BMMs. We could not observe any TRAP-positive multinucleated osteoclast-like cells on 2 day culture. We have added this result in Figure S2 and described this in the Results section (Lines 83-84).

  1. 3A, 3B, 3C.

It is curious that the number of osteoclasts generated by 20 nM S100P was smaller than that by 5 nM. The authors are recommended to show a dose-response curve of S100P on osteoclastogenesis.

>> As pointed out by the reviewer, we analyzed the dose-response (0, 5, 10, and 20 nM) effect of S100P on osteoclastogenesis. We also performed a dose-response effect of S100P to investigate a marker gene expression of osteoclast differentiation. These all results have been added to revised Figure 3 and the Results section (Lines 132-135 and 138-140).

The S100P you used (LSBio) appears to contain additives such as NP-40, NA3VO4, or 100 mM glycine and 10% glycerol or 10mM glutathione, etc. Such additives might enhance osteoclastogenesis. Did you remove the additives by dialysis or gel filtration? Or did you add the same amount of such additives to the control culture? If so, please describe it in “Materials and Methods”.

>> We apologize for this overlook. In this study, we did not remove the additives by dialysis or gel filtration. Human S100P protein (LSBio) was supplied in 25 mM Tris-HCl (pH 7.3), 100 mM glycine, and 10% glycerol. We experimented by adding the same amount of these additives to the control culture, and found that the additives did not alter osteoclastogenesis. We have described this information in the Results section (Lines 140-142).

  1. Line 162 - 163.  “…because the level of MK-expressing osteoprotegerin was high in MK CM, resulting in the inhibition of osteoclast development [26]”.

Reference 26 showed that the high levels of osteoprotegerin in MK-CM was not the factor that caused the inhibition of osteoclastogenesis. Did you neutralize or remove osteoprotegerin from your MK-CM by pretreatments? If not, please describe the possible effect of osteoprotegerin in your MK-CM in “Discussion”.

>> We appreciate this valuable comment. We performed microarray analysis to compare MKs to K562 cells using the Affymetrix GeneChip Human Gene 2.0 ST array. Of the 44,629 genes compared, a relative gene expression fold-change > 2, normalized data (log2) > 4, and p-value < 0.05, were considered to represent up- or down-regulation. However, the expression level of osteoprotegerin (TNFRSF11B, tumor necrosis factor receptor superfamily member 11 B) was exhibited fold-change of 0.945 and p-value of 0.399. This result indicates that osteoprotegerin expression was not altered in MKs compared to that in K562 cells in the culture system used in this study. We have included this information in the Discussion section (Lines 183-188).

Minor comments:

  1. Line 29. “demonstrating improved resorption activity”.

Line 142. “ bone resorption was promoted by S100P protein secreted from MKs”

The results in Figure 4 suggest that osteoclasts differentiated from mouse BMM on a dentine slice showed an increase in bone resorption in the presence of recombinant S100P. If you want to say “improve” or “promote”, you have to count the number of osteoclasts on a dentine slice and compare the ability of bone resorption such as resorption area/osteoclast. I think the results in Figure 4 merely indicate that the S100P enhanced the number of osteoclast-like cells that have the ability to resorb bone.

>> Thank you for this comment. We did not count the number of osteoclasts on dentin slice and compare resorption area/osteoclasts. Thus, as pointed out by the reviewer, we have revised the pointed sentence to “… demonstrating enhanced resorption activity” (Line 28) and “… was enhanced by S100P protein secreted from MKs” (Line 161) for explaining the results in Figure 4.

  1. The researcher in bone biology traditionally use the term “osteoclast-like cells” for the multinucleated cells differentiated in vitro. The term “osteoclasts” is only used for the cells isolated from bone. The manuscript would be improved if the author could follow the convention throughout the text.

>> Thank you very much for the kind comment. We have revised “osteoclasts” to “osteoclast-like cells” for multinucleated cells differentiated in vitro in the revised manuscript.

  1. The authors used the recombinant human S100P in their experiments. It will be useful for readers to explain why human recombinant worked on mouse BMM in “Discussion". Are there any differences in the protein sequence between human and mouse S100P?

>> S100 calcium-binding protein P (S100P) is encoded by S100P in humans. However, information on mouse S100P is insufficient. House mice possess S100P (BC038329.1) or a predicted gene, 40318 (Gm40318), as an S100P pseudogene. These genes have no nucleotide and protein sequence similarity to human S100P. Also, we were unable to obtain human osteoclasts or bone marrow macrophages. Therefore, we used mouse BMMs in this study. We have described this information in the Discussion section (Lines 220-224).

Reviewer 3 Report

Dear authors,

The authors examined the enhanced effect of the secreted molecule from megakaryocyte on osteoclastogenesis in vitro. They showed that megakaryocyte conditioned medium enhanced osteoclastogenesis induced by RANKL treatment in mouse BMM. Next, they focused on S100 protein form the data of microarray analysis. Finally, they found that S100 protein enhanced osteoclastogenesis. They concluded that megakaryocyte-derived S100 protein was responsible for enhanced osteoclastogenesis. However, this story was not adequately supported by the results. Some results should be confirmed by additional experiments.

Important issue

Although the authors used human leukemic cell line (K562) as a megakaryocyte model, the identities of S100P between human and mouse is not high as shown below. At least, recombinant S100P should be originated from mouse. How do the authors explain these species difference?

Major

Fig. 1 Control group was missing. The authors should show osteoclast differentiation without conditioned medium.

Table 1 The authors listed up the candidates of factor that enhance osteoclastogenesis. The authors should show only secreted molecules. Among them, the authors should explain the reason why the authors focused on only S100P.

Fig. 3 The authors showed the effect of recombinant S100P (rS100P)on osteoclastogenesis. Why did the authors use rS100P of human origin?

Fig. 3 Why did the authors use rS100P at concentrations of 5nM and 20nM for this experiment? According to Fig. 2, in my calculation, S100P secreted level from mature megakaryocytes was about 50 pM (500pg/ml).

Author Response

Reviewer 3

The authors examined the enhanced effect of the secreted molecule from megakaryocyte on osteoclastogenesis in vitro. They showed that megakaryocyte conditioned medium enhanced osteoclastogenesis induced by RANKL treatment in mouse BMM. Next, they focused on S100 protein form the data of microarray analysis. Finally, they found that S100 protein enhanced osteoclastogenesis. They concluded that megakaryocyte-derived S100 protein was responsible for enhanced osteoclastogenesis. However, this story was not adequately supported by the results. Some results should be confirmed by additional experiments.

Important issue

Although the authors used human leukemic cell line (K562) as a megakaryocyte model, the identities of S100P between human and mouse is not high as shown below. At least, recombinant S100P should be originated from mouse. How do the authors explain these species difference?

>> Thank you very much for raising this important issue. As a megakaryocyte model, we used a human leukemic cell line (K562) and selected S100P, which was secreted by MKs, as a candidate for increasing osteoclast differentiation. S100 calcium-binding protein P (S100P) is encoded by S100P gene in humans. However, information on mouse S100P is insufficient. House mice possess S100P (BC038329.1) or a predicted gene, 40318 (Gm40318), as an S100P pseudogene. These genes have no nucleotide or protein sequence similarity to human S100P. In addition, we were unable to obtain human osteoclasts or bone marrow macrophages. Finally, we used recombinant human S100P in mouse bone marrow macrophages. This is described in the Discussion section (Lines 220-224).

Major

Fig. 1 Control group was missing. The authors should show osteoclast differentiation without conditioned medium.

>> We apologize for not including the control group in the data shown in Figure 1. We repeated the experiment by adding a group without conditioned medium (non-CM) and have included this information in revised Figure 1 and figure legend (Lines 90-91).

Table 1 The authors listed up the candidates of factor that enhance osteoclastogenesis. The authors should show only secreted molecules. Among them, the authors should explain the reason why the authors focused on only S100P.

>> Thank you for raising this question. We apologize for the confusion regarding the description of Table 1. Table 1 lists the secretory genes that were upregulated in MKs compared to those in K562 cells. We have described this information in the Results section (Lines 103-104). Among the secretory genes shown in Table 1, S100P was selected as a candidate for enhancing osteoclastogenesis based on literature reviews. We have described this information in the Discussion section (Lines 204-214) with References.

Fig. 3 The authors showed the effect of recombinant S100P (rS100P) on osteoclastogenesis. Why did the authors use rS100P of human origin?

>> As described in response to the important issue raised by the reviewer, no information on mouse S100P was found. House mice possess S100P (BC038329.1) or a predicted gene, 40318 (Gm40318), as an S100P pseudogene, but they have no nucleotide or protein sequence similarity to human S100P. We have described this information in the Discussion section (Lines 220-224). Moreover, human recombinant S100P is commercially available.

Fig. 3 Why did the authors use rS100P at concentrations of 5nM and 20nM for this experiment? According to Fig. 2, in my calculation, S100P secreted level from mature megakaryocytes was about 50 pM (500pg/ml).

>> Thank you very much for the nice comment. The S100P level secreted from MKs was approximately 50 pM, as shown by ELISA in Figure 2, but we could not conclude that the levels were consistent with those in MKs. Because the amount of secreted S100P protein may vary slightly depending on the number or condition of MKs. Therefore, to better understand the role of S100P in osteoclastogenesis, we used a higher concentration than that secreted by MKs. We have mentioned this point in the Results section (Lines 129-132).

Round 2

Reviewer 3 Report

The reviewer asked the reason why the authors used human rS100P instead of mouse rS100P.

The author replied as follows;

  1. The information of mouse S100P is insufficient.
  2. It is difficult for the authors to use human monocyte-derived osteoclasts.

But, the reviewer cannot understand these reasons. Why do they suggest human S100P accelerates mouse osteoclast differentiation, although the information of mouse S100P is insufficient. Please describe scientifically the reason why the authors suggest human S100P affect osteoclastogenesis as mouse S100P.

The reviewer pointed out the concentration of rS100P used in this experiment.

The author replied as followes;

  1. The S100P level secreted from MKs was approximately 50 pM, as shown by ELISA in Figure 2, but we could not conclude that the levels were consistent with those in MKs. Because the amount of secreted S100P protein may vary slightly depending on the number or condition of MKs.
  2. To better understand the role of S100P in osteoclastogenesis, we used a higher concentration than that secreted by MKs.

The reviewer cannot agree with this reply, because the authors showed the conditioned medium of MK accelerated osteoclastogenesis in figure 1.Is there the difference between MK-CM in figure 1B and figure 2C. If so, the authors should measure the concentration of S100P in the condition of figure 1B. The reviewer think that the concentration of S100P should be at physiological conditions.